# Point Cluster: A Compact Message Unit for Communication-Efficient Collaborative Perception

**Zihan Ding**[1]**, Jiahui Fu**[1]**, Si Liu**[1]*****Hongyu Li**[1]**, Siheng Chen**[2]**,**
**Hongsheng Li**[3,4]**, Shifeng Zhang**[5]**, Xu Zhou**[5]
[1] Institute of Artificial Intelligence, Beihang University,
[2] School of Artificial Intelligence, Shanghai Jiao Tong University,
[3] MMLab, CUHK, [4] Centre for Perceptual and Interactive Intelligence, [5] Sangfor Technologies
`dingzihan737@gmail.com`

## Abstract

The objective of the collaborative perception task is to enhance the individual agent's perception capability through message communication among neighboring agents. A central challenge lies in optimizing the inherent trade-off between perception ability and communication cost. To tackle this bottleneck issue, we argue that a good message unit should encapsulate both semantic and structural information in a sparse format, a feature not present in prior approaches. In this paper, we innovatively propose a compact message unit, namely point cluster, whose core idea is to represent potential objects efficiently with explicitly decoupled low-level structure information and high-level semantic information. Building upon this new message unit, we propose a comprehensive framework **CPPC** for communication-efficient collaborative perception. The core principle of **CPPC** is twofold: first, through strategical point sampling, structure information can be well preserved with a few key points, which can significantly reduce communication cost; second, the sequence format of point clusters enables efficient message aggregation by set matching and merging, thereby eliminating unnecessary computation generated when aligning squared BEV maps, especially for long-range collaboration. To handle time latency and pose errors encountered in real-world scenarios, we also carefully design parameter-free solutions that can adapt to different noisy levels without finetuning. Experiments on serval widely recognized collaborative perception benchmarks showcase the superior performance of our method compared to the previous state-of-the-art approaches.

## 1 Introduction

Collaborative perception aims to enhance the perception capabilities of individual agents by utilizing complementary information exchanged between surrounding agents. This approach offers a novel strategy to address several inherent challenges of single-agent perception He et al. (2017); Cheng et al. (2022), including occlusion and long-range limitations. There is a pressing need for related methods and systems across a wide spectrum of practical applications, such as vehicle-to-everything autonomous driving Yu et al. (2022), automated multi-robot warehouse systems Li et al. (2020), and multi-UAVs for search and rescue Scherer et al. (2015). Recent efforts have made valuable contributions in terms of high-quality real and simulated datasets Yu et al. (2022); Xu et al. (2022b); Li et al. (2022); Xu et al. (2022a; 2023b), as well as effective solutions Xu et al. (2023a); Li et al. (2021a); Wang et al. (2020); Xu et al. (2022a); Hu et al. (2022); Chen et al. (2019a); Xu et al. (2022b) for collaborative perception.

The paramount challenge in this field involves optimizing perceptual ability in the face of limited communication bandwidth in real-world scenarios, i.e., communication-efficiency. Based on the type of observation transmission medium (i.e., message unit), previous studies can be categorized into three types: *early collaboration* with raw point cloud data Chen et al. (2019b); Arnold et al. (2020), *late collaboration* using bounding boxes Shi et al. (2022); Zeng et al. (2020); Rauch et al. (2012); Glaser & Kira (2023), and *intermediate collaboration* through bird's eye view (BEV) maps Xu et al. (2022b;a; 2023a); Hu et al. (2022). Early collaboration can simplify subsequent

---
*Corresponding author

analyses and enrich information for downstream models. Despite fostering high performance, this approach incurs huge bandwidth consumption due to the transmission of complete raw observations. Late collaboration ensures efficient economic bandwidth consumption; however, it suffers from performance bottlenecks and reduced robustness due to scarce object information. Intermediate collaboration compresses representative information into BEV feature maps, resulting in reduced communication bandwidth compared to early collaboration, while also enhancing perception capabilities in comparison to late collaboration. Despite significant development, the manual projection of point clouds to BEV feature maps suffers from quantization error Shi et al. (2019) and creates a bottleneck in collaboration, hindering these methods from effectively completing 3D structural information for precise object boundary predictions.

Considering the above issues, we innovatively propose a compact message unit, called *point cluster*, which describes objects with point coordinates representing the object structure, a cluster center representing the object position, and cluster feature representing the high-level semantics of the object. The point cluster has several merits compared to existing message units: 1) Unlike raw point clouds, point clusters intrinsically capture only the foreground objects within a scene in a sparse manner, leading to efficient communication; 2) In contrast to bounding boxes, point clusters offer sufficient information about potential objects, enabling more robust aggregation and decoding for enhanced regression accuracy; 3) Compared to BEV maps, point clusters explicitly preserve the structural information of objects in original coordinate space, enabling fine-grained point alignment and complementary structure information fusion between different agents.

To fully unleash the potential of this powerful message unit, we propose **CPPC**, a comprehensive **C**ollaborative **P**erception framework based on **P**oint **C**luster that revolutionizes existing communication mechanism, while featuring low bandwidth usage, high-performance perception, and real-noise robustness. The main body of our **CPPC** includes two key modules: 1) a point cluster packing module, which flexibly controls the number of points contained in each point cluster while maintaining their geometric structure to cope with different bandwidth constraints; 2) a point cluster aggregation module, which integrates point clusters from other agents with set matching and merging to achieve a comprehensive understanding of the surrounding scene for the ego agent. In addition to optimizing bandwidth and performance trade-off, our **CPPC** also has careful designs for two common robustness challenges, i.e., pose error Lu et al. (2023) and time delay Lei et al. (2022); Wei et al. (2024), benefited from the low-level information in point cluster. To explicitly solve the pose error, we propose to reformulate cluster centers as vertices of a graph and optimize to promote pose consistency between agents and point clusters. In order to compensate for the time latency, we link point clusters of the same object along the time dimension and measure its speed for position prediction in the current timestamp.

As an intermediate collaboration method, using point cluster as message unit gives our **CPPC** three distinct advantages compared with previous methods based on BEV map: 1) Decoupling of structure and semantic information allows our **CPPC** to efficiently represent each potential object with a few key points and significantly fewer feature channels, all while avoiding the loss of high-level information during message packing. 2) The computational complexity of our **CPPC** during message aggregation is efficiently related to the number of possible objects in the scene, rather than quadratically related to the perception range, which is more suitable for long-range collaboration. 3) Our graph optimization and speed estimation methods possess a strong generalization capability as they do not rely on any training parameters during the optimization process. This enables our method to adapt to varying levels of noise with ease. To validate the effectiveness of our proposed **CPPC**, we conduct experiments on three popular collaborative perception benchmarks, V2X-Set Xu et al. (2022a), OPV2V Xu et al. (2022b), and DAIR-V2X Yu et al. (2022). Our method achieves the new state-of-the-art performances with significant performance gains of 5.7%, 7.3% and 12.8% on the strict mAP@0.7 metric, respectively. Not only that, we also contributes this community a series of new metrics to demonstrate the benefits of different message units brings across various collaboration phases in a fine-grained manner, where our **CPPC** still achieves the best performance.

## 2 PROBLEM FORMULATION

Suppose there are $N_{\text{agent}}$ agents present in the scene, whose observations and the ground truth annotation are denoted as $\{\mathcal{X}^i\}_{i=1}^N$ and $\{\mathcal{Y}^i\}_{i=1}^N$, respectively. Collaborative perception aims to maximize the perception performance of all agents while taking into account the constraint on available

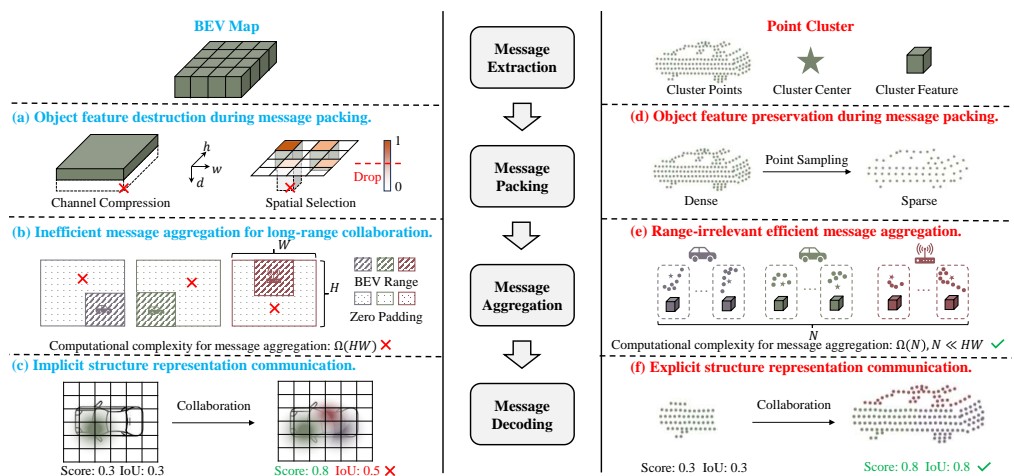

Figure 1: Illustration of using BEV map and point cluster as message units for intermediate collaboration.

bandwidth $\beta$, which can be formulated as:

$$\arg\max_{\theta,\mathcal{M}} \sum_{i=1}^{N} g(\Phi_\theta(\mathcal{X}^{i;t}, \{\mathcal{M}^{j;t-\tau^{j\to i;t}}\}_{j=1,j\neq i}^{N}), \mathcal{Y}^{i;t}), \text{s.t.} \sum_{i=1}^{N}\sum_{j=1,j\neq i}^{N} |\mathcal{M}^{j;t-\tau^{j\to i;t}}| \leq \beta. \quad (1)$$

Here, $\theta$ denotes trainable parameters of the network $\Phi$ and $g(\cdot,\cdot)$ is the evaluation metric. $\mathcal{M}^{j;t-\tau^{j\to i;t}}$ is the message transmitted from the $j$-th agent at the timestamp $t - \tau^{j\to i;t}$ and received by the $i$-th agent at the timestamp $t$, where $\tau^{j\to i;t}$ is the transmission latency. It includes the $j$-th agent's extracted collaborative message units $\boldsymbol{M}^j$ and 6DoF pose $\xi^j$. Note that 1) there is no collaboration when $\beta = 0$ and the objective reflects the single-agent perception performance; 2) when accounting for pose error, it is necessary to correct the pose $\xi^j$ for feature alignment; 3) we will omit the superscripts $t$ related to the timestamp in the following when $\tau = 0$ for simplicity.

Existing intermediate collaboration approaches employing dense BEV maps as basic message units have several limitations: 1) *Object feature destruction during message packing* (Figure 1 (a)). Channel compression Xu et al. (2022a) enables the sender to preserve spatially complete scene information, while the receiver may suffer from object feature degradation during reconstruction due to heterogeneity representation across channels. In contrast, spatial selection Hu et al. (2022); Wang et al. (2023b) transmits only informative regions pointed out by spatial confidence maps, which may result in potential object loss when bandwidth constraints become more stringent. 2) *Inefficient message aggregation for long-range collaboration* (Figure 1 (b)). Collaboration can bring a larger perception range to the ego agent, but computational complexity also grows quadratically with the expansion of dense BEV feature maps. Moreover, limited by the square map structure and convolution operation requirements, the received BEV features are inevitably filled to the same shape for aggregation with zero paddings, resulting in unnecessary calculation on non-overlapped areas. 3) *Implicit structure representation communication* (Figure 1 (c)). The voxelization operation sacrifices 3D geometric details in comparison to the raw point clouds. While aggregating BEV representations from different agents can enhance the response of potential object regions, the precision of predicted box boundaries may be constrained by incomplete object structure modeling.

Our **CPPC** based on point clusters for communication can overcome these issues: 1) *Object feature preservation during message packing.* (Figure 1 (d)). Point clusters inherently contain only the information of foreground objects present in the scene, eliminating the need for filtering out irrelevant backgrounds by handcrafted rules. Moreover, we can control the transmission bandwidth by explicitly reducing the number of points, as opposed to implicitly compressing feature channels. 2) *Range-irrelevant efficient message aggregation* (Figure 1 (e)). The number of point clusters is more related to the number of objects in the scene rather than the collaboration range. Furthermore, point clusters can be conveniently associated with the same object and aggregated through set merging, without the need for padding to the same shape of joint field of view. 3) *Explicit structure representation communication* ((Figure 1 (f))). Point clusters fully preserve the geometric structural information of objects in the original coordinate space, enabling fine-grained point alignment and complementary structure information fusion between different agents, which can improve the precision of predictions.

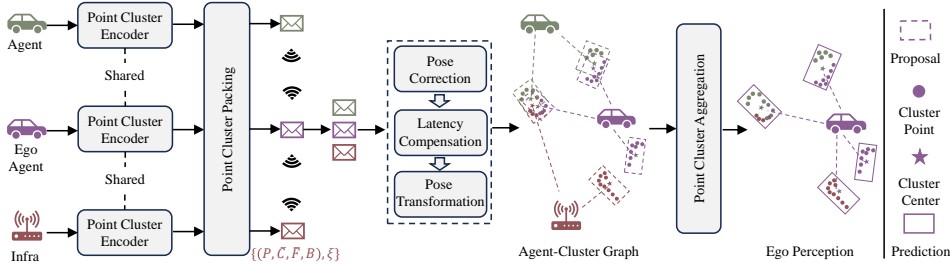

Figure 2: Overview of our **CPPC** system, including a shared **PCE** module to extract point clusters from raw point clouds, a **PCP** module to selectively packing informative point clusters, and a **PCA** to complete object information. Pose error and time latency are revised before aggregation phase.

## 3 METHOD

To begin with, we will present an overview of our **CPPC** framework in §3.1. The we describe details about how we encode, pack, and aggregate point clusters in §3.3 and §3.4, respectively. Finally, we will address time latency and pose error problems to improve our method's robustness in §3.5.

### 3.1 OVERVIEW

Figure 2 illustrates the overall architecture of our proposed method. Initially, the raw point clouds of all agents are processed by a shared Point Cluster Encoder (**PCE**), which segments foreground points on the surface of objects and divides them into clusters based on the distance metric. From each of these point clusters, we extract and formulate the point coordinates, the center coordinates, and the cluster feature as corresponding intermediate representations. Then we propose a Point Cluster Packing (**PCP**) module to filter noisy background clusters and correct the involved points of foreground clusters via proposal generation. The reduction in bandwidth usage of the point cluster can be achieved by decreasing the number of included points. After receiving messages from other agents, we address the pose error and time latency with parameter-free approaches and then align the coordinate space of point clusters from multiple agents via pose transformation, which allows the ego agent to obtain an agent-cluster graph as the comprehensive scene representation from its own coordinate space. As for message aggregation, we propose a Point Cluster Aggregation (**PCA**) module, where point cluster matching is performed to find point clusters belonging to the same object and merge them into a new point cluster that contains complete object information with linear complexity. Finally, we refine cluster features of point clusters via point-based operators, based on which we output the final detection results.

### 3.2 POINT CLUSTER ENCODER

Since the original BEV representations for collaborative perception suffer from object feature destruction, inefficient message aggregation for long-range collaboration, and implicit structure representation communication, we propose a brand new representation called point cluster to address these issues. We adopt the encoder-decoder point cloud backbone Shi et al. (2020b) based on 3D sparse convolution and deconvolution as the sparse voxel feature extractor. To construct each point feature, we concatenate the voxel feature where the point is located and the corresponding offset from the point to the voxel center. These point features are then passed through a MLP for foreground segmentation. Since objects are naturally well-separated, 3D box annotations in autonomous driving scenes directly provide semantic masks for supervision Shi et al. (2019). We use focal loss Lin et al. (2017) for segmentation loss, denoted as $\mathcal{L}_{\text{seg}}$. For foreground points, we use another MLP to predict their offsets to the corresponding object centers, which is supervised by L1 loss Ren et al. (2015), denoted as $\mathcal{L}_{\text{center}}$. Next, we measure the distance among the predicted centers of foreground points, where two points belong to the same point cluster if their predicted centers' Euclidean distance is smaller than a certain threshold $\epsilon_{\text{point}}$.

After we can directly extract cluster features via point-based operators, such as PointNet Qi et al. (2017a), DGCNN Wang et al. (2019), Meta-Kernel Fan et al. (2021), and SIR Fan et al. (2022). We select SIR in this paper and stack $L_1$ layers to encode all cluster features in parallel. To express more clearly, we take the processing process of the $q$-th cluster in the $l$-th SIR layer as an example. Concretely, assume there are $N_{\text{point}}^q$ foreground points, we denote the included point coordinates and features as $\boldsymbol{P}^q \in \mathbb{R}^{N_{\text{point}}^q \times 3}$ and $\boldsymbol{F}_{\text{point}}^{q;l} \in \mathbb{R}^{N_{\text{point}}^q \times D}$, respectively, where $D$ is number of feature channels. We take the average coordinates of all predicted cluster centers as the cluster center,

denoted as $C^q \in \mathbb{R}^{1 \times 3}$. The processing process of SIR can be formulated as:

$$\widetilde{F}_{\text{point}}^{q;l} = \text{MLP}([F_{\text{point}}^{q;l}; P^q \ominus C^q]), F_{\text{point}}^{k;l+1} = \text{MLP}([\widetilde{F}_{\text{point}}^{q;l}; \text{maxpool}(\widetilde{F}_{\text{point}}^{q;l})]), \tag{2}$$

where $[;]$ denotes concatenation along the channel dimension, $\ominus$ means applying subtraction on each point in $P^q$, and $F_{\text{point}}^{q;l+1} \in \mathbb{R}^{N_{\text{point}}^q \times D}$ is the processed point cluster feature. We concatenate $\{F_{\text{point}}^{q;l}\}_{l=1}^{L_1}$ along the channel dimension, and apply linear transformation and max-pooling on it to obtain the final cluster feature $F^q \in \mathbb{R}^{1 \times D}$.

## 3.3 POINT CLUSTER PACKING

The intermediate representations of point clusters extracted from `PCE` by the $i$-th agent can be formulated as:

$$M^i = \{m^{i;q}\}_{q=1}^{N_{\text{cluster}}^i} = \{(P^{i;q}, C^{i;q}, F^{i;q})\}_{k=1}^{N_{\text{cluster}}^i}, \tag{3}$$

where $m^{i;q}$ is the representation of the $q$-th cluster, $N_{\text{cluster}}^i$ is the number of point clusters extracted by the $i$-th agent, $P^{i;q}$ denotes the set of included point coordinates, $C^{i;q}$ denotes the cluster center, and $F^{i;q}$ denotes the cluster feature. However, during the point cluster grouping process, errors may occur, resulting in the loss of points on the object's surface and the inclusion of background distractors. In order to correct this issue, we propose to generate a proposal bounding box for each cluster. To achieve this, we feed the cluster features to two separate MLPs for proposal classification and regression. During the training phase, clusters are classified as positive if their predicted centers are located in the ground truth bounding boxes. We adopt L1 loss Ren et al. (2015) and focal loss Lin et al. (2017) as regression loss $\mathcal{L}_{\text{reg}}$ and classification loss $\mathcal{L}_{\text{cls}}$, respectively. After proposal generation, we only retain point clusters with positive proposals. The point coordinates in these clusters are overridden with the coordinates of points within the positive proposals. Consider there are $N_{\text{cluster+}}^i$ positive clusters, we modify the formulation of $M_i$ as following:

$$M^i = \{m^{i;q}\}_{q=1}^{N_{\text{cluster+}}^i} = \{(P^{i;q}, C^{i;q}, F^{i;q}, B^{i;q})\}_{q=1}^{N_{\text{cluster+}}^i}, \tag{4}$$

where $B^{i;q} = (\hat{x}, \hat{y}, \hat{z}, \hat{h}, \hat{w}, \hat{l}, \alpha, \hat{c})$ includes the center coordinates, the size, the yaw angle and the confidence score of the proposal bounding box of the $q$-th cluster. The final message distributed can be formulated as $\mathcal{M}^i = (M^i, \xi^i)$, where $\xi^i$ is the 6DoF pose used for pose transforming in the later aggregation phase.

Optimizing the trade-off between perception performance and communication bandwidth is vital in collaborative perception. Unlike BEV maps, our point clusters are inherently sparse in the spatial dimensions. Therefore, the communication cost in our method is mainly on the point coordinates (*i.e.* thousands of 3-dimensional coordinates) rather than the object features (*i.e.* one 128-dimensional features). Considering that the geometric structure of the object can be represented by keypoints, we propose **S**emtanic and **D**istribution guided **F**arthest **P**oint **S**ampling (`SD-FPS`) to compress the transmission data, which can effectively exclude raw points with ambiguous semantic features or redundant structural features. Specifically, not only considering the distance $d_{\text{point}}$ between points like naive farthest point sampling, but also we select keypoints based on the object's semantic confidence score $s_{\text{f}}$ and distribution density score $s_{\text{d}}$. The semantic confidence score is derived from segmentation head in `PCE`, with a higher score indicating richer semantic information for distinguishing objects. The distribution density score for point $p$ is inversely proportional to its density estimation: $\frac{1}{|\mathcal{N}(p)|} \sum_{q \in \mathcal{N}(p)} K(p, q)$, where $\mathcal{N}(p)$ is a point set in the nearby area of $p$, and $K(\cdot, \cdot)$ is a gaussian kernel function to measure the similarity between position of two points. Thus, a lower $s_{\text{d}}$ indicates removing it will not significantly affect the object's shape. We show the algorithm details of applying `SD-FPS` on the $q$-th cluster of the $i$-th agent in Appendix A.1.

## 3.4 POINT CLUSTER AGGREGATION

After message communication, we need to appropriately aggregate point clusters from surrounding agents to form a holistic perception. We take the aggregation process from the $j$-th agent to the $i$-th agent as an example, which can be extended to all agents easily. Firstly, we align the coordinate space of $M^j$ to that of $M^i$ through the transform matrix calculated from $\xi^i$ and $\xi^j$. The transformed message units from the $j$-th agent are denoted as $M^{j \to i}$. Similar to the foreground point grouping process, we match $M^i$ and $M^{j \to i}$ based on their clusters' centers, where the $q$-th point cluster of the $i$-th agent $m^{i;q}$ and the $r$-th point cluster of the $j$-th agent $m^{j \to i;r}$ belong to the same object if

their centers' distance $\|C^{i;q} - C^{j \to i;r}\|_2$ is less than a predefined threshold $\epsilon_{\text{agg}}$. After matching, we organize all point clusters as two disjoint sets $M_{\text{unique}}$ and $M_{\text{share}}$. The set $M_{\text{unique}}$ comprises point clusters exclusively observed by a single agent, which do not need to be involved in the following aggregation process. Differently, $M_{\text{share}}$ is the set of tuples including point clusters belonging to the same object in the scene. We combine each tuple to form a novel point cluster that encompasses comprehensive low-level and high-level object information. In detail, assume that $m^{i;q}$ and $m^{j \to i;r}$ belong to the $s$-th object in the scene, the aggregated point cluster $\ddot{m}^s = (\ddot{P}^s, \ddot{C}^s, \ddot{F}^s, \ddot{B}^s)$ can be formulated as follows:

$$\ddot{P}^s = P^{i;q} \cup P^{j \to i;r}, \quad \ddot{C}^s = \frac{C^{i;q} + C^{j \to i;r}}{2}, \tag{5}$$

$$\ddot{F}^s = \text{avgpool}(F^{i;q}, F^{j \to i;r}), \quad \ddot{B}^s = \begin{cases} B^{i;q}, & \text{if } \hat{c}^{i;q} - \hat{c}^{j \to i;r} \geq 0 \\ B^{j \to i;r}, & \text{otherwise} \end{cases}, \tag{6}$$

Since there are no convolution operations and unnecessary zero-padding, the computational complexity of aggregating point clusters in our `PCA` is only strongly related to the number of potential objects, which is more efficient for long-range collaboration than BEV-based aggregation methods.

### 3.5 ROBUSTNESS

In realistic communication settings, the presence of pose error and time latency is unavoidable and it leads to misalignment of point clouds that significantly affect the reliability of transferred information in collaborative perception. Benefiting from the low-level object information in point clusters, we propose parameter-free approaches that can generalize to different noise settings. For simplicity, we take the correction processes that happen between the $i$-th (ego) and $j$-th agents as an example.

**Pose Correction.** To address this issue, we propose to align the clusters from different agents belonging to the same object. After receiving the message from the $j$-th agent, we first align the coordinate space of $M^j$ to the $i$-th agent with relative pose $\xi^{j \to i} = (\xi^i)^{-1} \circ \xi^j$, where $\circ$ means multiplying their homogeneous transformation matrices. We denote $m^{i;q}$ and $m^{j \to i;r}$ as point clusters belonging to the $s$-th unique object in the scene after spatial cluster matching with threshold $\epsilon_{\text{pose}}$. In the following, we simplify each pose in 2D space. The pose of the $s$-th unique object is defined as $\chi^s = \xi^i \circ (C^{i;q} + C^{j \to i;r})/2$. Inspired by CoAlign Lu et al. (2023), we define the pose consistency error vector as $e^{js} = C^{j;r} \circ ((\xi^j)^{-1} \circ \chi^s)$, which is zero when there is no pose error. The overall optimization problem can be formatted as follows:

$$\{(\chi^s)', (\xi^j)'\} = \underset{\{\xi^j, \chi^s\}}{\arg \min} \sum_{j=1}^{N_{\text{agent}}} \sum_{s=1}^{N_{\text{object}}} (e^{js})^T e^{js}. \tag{7}$$

Different from existing BEV-based methods that need additional detection results for pose correction, we can directly utilize low-level information contained in the point cluster.

**Latency Compensation.** SyncNet Lei et al. (2022) proposes to complete BEV maps at the current timestamp using locally stored historical BEV maps from other agents. Differently, we propose to directly predict the location of point clusters in the current timestamp via speed estimation based on the low-level coordinate information, which is more efficient and interpretable. We denote the received point clusters from the $j$-th agent at the $t$-th timestamp as $M^{j;t-\tau^{j \to i;t}}$, where $\tau^{j \to i;t}$ is the time latency. The stored point clusters from the $j$-th agent during the last communication round are denoted as $M^{j;t'}$, where $t' < t - \tau^{j \to i;t}$. Similar to the spatial matching process in 3.4, we match point clusters along the time dimension by measuring whether their Euclidean distance is in a predefined range $[\underline{\epsilon}_{\text{latency}}, \overline{\epsilon}_{\text{latency}}]$. Assume $m^{j;r;t'}$ and $m^{j;q;t-\tau^{j \to i;t}}$ belong to the same object, we can infer its speed $v^{j;q;t-\tau^{j \to i;t}}$ and offset $\Delta d_{\text{cluster}}^{j;q;t-\tau^{j \to i;t}}$ to the current timestamp $t$ as follows:

$$v^{j;q;t-\tau^{j \to i;t}} = \frac{\|C^{j;q;t-\tau^{j \to i;t}} - C^{j;r;t'}\|_2}{t - \tau^{j \to i;t} - t'}, \Delta d_{\text{cluster}}^{j;q;t-\tau^{j \to i;t}} = v^{j;q;t-\tau^{j \to i;t}} \times \tau^{j \to i;t}. \tag{8}$$

Finally, we can obtain the estimated point cluster $m^{j;q;t}$ at the $t$-th timestamp by coordinates translation with $\Delta d_{\text{cluster}}^{j;q;t-\tau^{j \to i;t}}$.

# 4 EXPERIMENTS

## 4.1 DATASETS AND EVALUATION METRICS

We conducted experiments on three widely used benchmarks for collaborative perception, i.e., V2XSet Xu et al. (2022a), OPV2V Xu et al. (2022b), and DAIR-V2X-C Yu et al. (2022). We follow previous works Hu et al. (2022); Xu et al. (2022a) to select one of the agents in the scene as the ego agent, whose detection results are assessed by Average Precision (AP) at Intersection-over-Union (IoU) thresholds of 0.5 and 0.7, denoted as AP@0.5 and AP@0.7. Nevertheless, they treat all elements in the scenario uniformly, disregarding the input of collaborative participants. To compare the collaborative perception ability of different methods in a more fine-grained manner, we calculate the number of points observed by the ego agent $N_{ego}$ in all target objects and categorize them as: 1) **S**ingle-agent **P**erception of **O**ther agents (SP-O), which signifies objects that are scarcely perceived by the ego agent; 2) **S**ingle-agent **P**erception of **E**go agent (SP-E), which denotes objects that are more effectively scanned by the individual agent, with additional information provided by other agents primarily serving as supplementary; 3) **C**ollaborative **P**erception (CP), which indicates objects that are partially observable by the individual agent and require assistance from other agents to obtain comprehensive information about the objects. We evaluate the AP@0.7 of each group, denoted as $AP_{SP-O}$, $AP_{CP}$, and $AP_{SP-E}$, respectively.

## 4.2 IMPLEMENTATION DETAILS

We set the perception range along the $x$, $y$, and $z$-axis to $[-140.8m, 140.8m] \times [-40m, 40m] \times [-3m, 1m]$ for V2XSet and $[-100.8m, 100.8m] \times [-40m, 40m] \times [-3m, 1m]$ for DAIR-V2X-C, respectively. The communication results measure message size in bytes using a logarithmic scale with base 2. The thresholds $\epsilon_{agg}$, $\epsilon_{pose}$, $\underline{\epsilon}_{latency}$, and $\bar{\epsilon}_{latency}$ for cluster matching are set as 0.6, 1.5, 0.5 and 2.0, respectively. The number of SIR layers is $L_1 = 6$ in PCE and $L_2 = 3$ during message decoding. The channel number of cluster features is $D = 128$. Adam Kingma & Ba (2014) is employed as the optimizer for training our model end-to-end on NVIDIA Tesla V100 GPUs, with a total of 35 epochs. The initial learning rate is set as 0.001 and we reduce it by 10 after 20 and 30 epochs, respectively. Our method is implemented with PyTorch.

Table 1: Comparison with state-of-the-art methods on the test sets of V2XSet, OPV2V, and DAIR-V2X-C on perfect setting. The second highest performance accuracy is highlighted in blue in the table.

Table 2: Comparison with state-of-the-art methods on the test set of DAIR-V2X-C with different target categories.

| Method | V2XSet | | OPV2V | | DAIR-V2X-C | |
|---|---|---|---|---|---|---|
| | AP@0.5 | AP@0.7 | AP@0.5 | AP@0.7 | AP@0.5 | AP@0.7 |
| DiscoNet Li et al. (2021a) | 90.78 | 83.81 | 86.2 | 73.3 | 69.28 | 58.56 |
| V2X-ViT Xu et al. (2022a) | 89.03 | 79.02 | 91.4 | 82.3 | 73.98 | 61.50 |
| Where2comm Hu et al. (2022) | 85.03 | 76.77 | - | - | 71.48 | 60.36 |
| OPV2V Xu et al. (2022b) | 91.88 | 84.75 | 89.9 | 76.5 | 66.07 | 50.92 |
| CoBEVT Xu et al. (2023a) | 90.33 | 82.69 | - | - | 63.90 | 51.67 |
| CoAlign Lu et al. (2023) | - | - | 94.5 | 86.8 | 74.60 | 60.40 |
| Ours | **92.83** | **89.55** | **94.6** | **93.1** | **76.89** | **69.39** |

| Method | Metric | | |
|---|---|---|---|
| | $AP_{SP-O}$ | $AP_{CP}$ | $AP_{SP-E}$ |
| Where2comm | 38.54 | 73.17 | 67.18 |
| V2X-ViT | 36.85 | 74.94 | 67.27 |
| Ours | **40.13** | **82.63** | **76.72** |

## 4.3 COMPARISON WITH STATE-OF-THE-ART METHODS

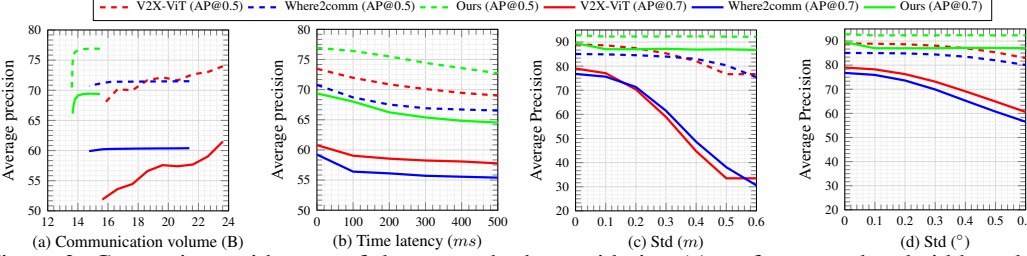

Figure 3: Comparison with state-of-the art methods considering (a) performance-bandwidth trade-off, (b) time latency, (c) positional error, and (d) heading error.

As shown in Table 1, our **CPPC** outperforms previous BEV-based approaches on the test sets of all evaluated benchmarks, indicating the effectiveness of adopting point cluster as the basic collaborative message unit in both simulated and realistic scenarios. Our **CPPC** outperforms other methods significantly in the most stringent metric AP@0.7, and is 5.7%, 7.3%, and 12.8% higher than the previous best OPV2V, CoAlign, and V2X-ViT on V2XSet, OPV2V, and DAIR-V2X-C, respectively, demonstrating its ability to accurately locate the correct object through collaborative perception and regress a more precise bounding box to cover the object. In addition, we compare our point cluster

aggregation method with late collaboration, which directly use $\ddot{B}^s$ for point clusters matching the same potential objects, i.e., in $M_{\text{share}}$, and $B$ for objects exclusively observed by a single agent, i.e., in $M_{\text{unique}}$, as the final outputs, respectively. Our method improves late collaboration by 15.0% and 68.2% on V2X-Set and DAIR-V2X on AP@0.7. This finding confirms that our `PCA` module can effectively utilize rich object information—specifically, semantic features and structural information—in point clusters received from surrounding agents, enabling the discovery of objects that cannot be accurately detected by individual agents.

We explore the performance-bandwidth trade-off of our `CPPC` and previous typical methods in Figure 3 (a). Thanks to the sparse nature of point clusters, the communication volume is under 16 even if we pack all cluster points, which is close to the lower bound of bandwidth usage of other methods. Under the same communication volume, our method achieves significant performance improvements of around absolute 10.0 AP@0.7. The little performance drops with bandwidth decrease indicate that appropriate point sampling strategies can reduce representation redundancy.

Time latency poses a pervasive challenge in real-world V2X communication, leading to asynchronization between ego features and received collaborative features. We compare the model robustness against time latency ranging from 0 to $500ms$ in Figure 3 (b). Compared to previous intermediate methods, our `CPPC` achieves superior performance, with similar AP degradation as latency increases. It is worth noting that both V2X-ViT and Where2comm need to be finetuned with data on different noisy levels, while our `CPPC` can adapt to arbitrary noise levels in a zero-shot manner.

Collaborative agents depend on precise pose data from others to transform coordinates of received messages. Despite advanced localization technologies like GPS, pose error is unavoidable. Therefore, collaborative approaches need to be resilient to localization errors. We compare different methods with pose noises following Gaussian distribution with standard deviations from $\{0.0, 0.1, 0.2, 0.3, 0.4, 0.5, 0.6\}$ for positional error $(m)$ (Figure 3 (c)) and heading error $(°)$ (Figure 3 (d)), respectively. Results show that there is no significant AP drop, which validates that our `CPPC` can handle large noise disturbances without further finetuning and additional model parameters.

The existing evaluation metrics in collaborative perception treat all objects equally, which is not conducive to fine-grained analysis. As shown in Table 2, we split targets into different categories based on the number of points scanned by the ego agent and evaluate AP@0.7 for each category, denoted as $AP_{\text{SP-O}}$, $AP_{\text{CP}}$, and $AP_{\text{SP-E}}$. Experimental results demonstrate the superiority of our `CPPC` over the state-of-the-art BEV-based approaches across all evaluation metrics. In detail, high $AP_{\text{SP-O}}$ means that our PCP module can keep more complete object information since we avoid feature destruction caused by channel compression and spatial selection. The results in the 2-nd column demonstrate the significant improvement of our `CPPC` over BEV-based methods in terms of the $AP_{\text{CP}}$ metric. This indicates that utilizing point clusters as the basic collaborative message unit is advantageous for both message packing and aggregation phases, resulting in enhanced collaboration compared to earlier techniques relying on dense BEV maps.

Table 3: Different numbers of feature channels in `PCE`.

| Metric | Channel Number | | | | | |
|--------|------|------|------|------|------|------|
|        | 8    | 16   | 32   | 64   | 96   | 128  |
| AP@50  | 74.41 | 75.32 | 77.12 | 76.63 | 77.40 | 77.30 |
| AP@70  | 64.79 | 67.08 | 69.32 | 68.92 | 69.55 | 69.43 |

### 4.4 ABLATION STUDIES

**Number of Feature Channels in `PCE`.** We assess AP@50 and AP@70 using various channel numbers of cluster features in `PCE` on the test set of DAIR-V2X-C in Table 3. The results indicate that there is no notable decrease in performance when the number of channels is reduced to 16. We argued that since we explicitly include structure representation in point clusters, the semantic information can be compressed to a large extent, leading to small bandwidth consumption. In contrast to prior approaches that compress solely before the message packing phase, our framework incorporates a small number of channels throughout the entire encoding phase, thereby decreasing network size and computational overhead without destroying object features across channels (3-rd and 4-st rows).

Table 4: Different sampling ratios and methods.

| Method | Ratio | | | | | |
|--------|-------|-------|-------|-------|------|------|
|        | 1/128 | 1/64  | 1/32  | 1/16  | 1/8  | 1/4  |
| RPS    | 64.60 | 65.46 | 67.12 | 67.80 | 68.61 | 68.75 |
| FPS    | 65.39 | 66.32 | 67.43 | 68.52 | 69.14 | 69.36 |
| S-FPS  | 66.21 | 67.08 | 68.04 | 69.01 | 69.05 | 68.96 |
| D-FPS  | 65.65 | 66.52 | 67.51 | 68.57 | 69.21 | 69.46 |
| SD-FPS | 66.12 | 67.03 | 68.08 | 69.22 | 69.19 | 69.41 |

**Sampling Point Clusters with Different Ratios and Methods.** We evaluate AP@70 of different sampling ratios and methods during message packing on the test set of DAIR-V2X-C in Table 4. The ratio is the number of sample points divided by the total number of points. The first and second

Table 5: Ablation experiments on $\epsilon_{\text{point}}$, $\epsilon_{\text{agg}}$, $\epsilon_{\text{pose}}$, and $\bar{\epsilon}_{\text{latency}}$.

| | $\epsilon_{\text{point}}$ | | | | | $\epsilon_{\text{agg}}$ | | | | | $\epsilon_{\text{pose}}$ | | | $\bar{\epsilon}_{\text{latency}}$ | | |
| | 0.1 | 0.2 | 0.3 | 0.4 | 0.5 | 0.3 | 0.4 | 0.5 | 0.6 | 0.7 | 1.0 | 1.5 | 2.0 | 1.5 | 2.0 | 2.5 |
|---|---|---|---|---|---|---|---|---|---|---|---|---|---|---|---|---|
| AP@0.5 | 91.94 | **92.01** | 91.92 | 91.75 | 91.47 | 91.97 | 91.95 | 91.99 | **92.01** | 91.97 | 89.86 | **90.00** | 89.80 | 72.50 | **72.69** | 72.68 |
| AP@0.7 | 89.88 | **89.99** | 89.78 | 89.73 | 89.51 | 89.79 | 89.86 | 89.88 | **89.99** | 89.95 | 86.63 | **87.37** | 86.93 | 64.44 | **64.61** | 64.59 |

rows represent the baseline methods of random sampling and basic FPS. The second, third, and fourth rows represent the results of introducing the semantic score, density score, and their joint application. It can be seen that the semantic score performs better than the baseline under smaller sampling ratios, because it preserves the semantic of object categories in extremely sparse structure. The density score performs better than the baseline under larger sampling ratios, because it can remove local redundant information. By combining the two scores, our method has significant performance advantages over the baseline at all sampling ratios. Moreover, `SD-FPS` does not incur significant time costs, less than 3% of time consumption in entire pipeline (Details in Apendix C.4). The reason is that our method applies `SD-FPS` to each point cluster, which contains limited object-level points (about average $10^2$ points in the V2X-Set dataset) rather than large-scale scene-level points (about average $10^4$ points in the V2X-Set dataset).

**Ablation on Hyperparameters** We evaluate AP@0.5/AP@0.7 of different lower bounds $\underline{\epsilon}_{\text{latency}}$ for matching during latency compensation in Table 6. In line with Figure 3 (b), an increase in time latency may result in performance degradation due to notable position shifts that complicate temporal alignment. If our method encounters relatively high time la-

Table 6: Lower bound for matching during latency compensation.

| $\underline{\epsilon}_{\text{latency}}$ | Time Latency | | | | |
| | 100 | 200 | 300 | 400 | 500 |
|---|---|---|---|---|---|
| 0 | 76.36/67.97 | 75.23/66.08 | 74.13/64.61 | 73.19/63.57 | 71.84/62.79 |
| 0.1 | 76.34/67.93 | 75.20/66.09 | 74.15/64.60 | 73.20/63.53 | 71.82/62.80 |
| 0.2 | 76.34/67.88 | 75.21/66.06 | 74.13/64.59 | 73.17/63.46 | 72.37/64.00 |
| 0.3 | 76.34/67.86 | 75.19/66.00 | 74.14/64.60 | 73.19/63.50 | 72.72/64.46 |
| 0.4 | 76.42/68.01 | 75.26/66.05 | 74.11/64.58 | 73.50/64.42 | 72.76/64.47 |
| 0.5 | 76.44/68.01 | 75.35/65.54 | 74.36/65.18 | 73.79/64.82 | 72.75/64.48 |

tency (e.g., $300ms$, $400ms$, and $500ms$), performance significantly decreases when $\underline{\epsilon}_{\text{latency}} = 0$ or is very low. Our research revealed that certain vehicles remain stationary, rendering the assumption of uniform motion in Section 3.5 invalid. By adjusting $\underline{\epsilon}_{\text{latency}}$ as demonstrated in the 3-rd, 4-th, and 5-th columns, we can mitigate this issue by filtering out stationary targets during latency compensation. Other involved hyperparameters i.e., $\epsilon_{\text{point}}$, $\epsilon_{\text{agg}}$, $\epsilon_{\text{pose}}$, and $\bar{\epsilon}_{\text{latency}}$, in our **CPPC** are evaluated with different values in Table 5.

### 4.5 QUALITATIVE ANALYSIS

As shown in Figure 4, we show qualitative evaluation of our **CPPC** w/ and w/o collaboration on the test sets of both V2XSet and DAIR-V2X-C datasets. We can observe that the absence of collaboration leads to the overlooking of many objects to the ego agent's inability to perceive sufficient information regarding long-range or obscured targets. These issue can be effectively addressed using our message packing and aggregation mechanisms based on point clusters.

We also visualize the detection results before and after latency compensation on the test set of the DAIR-V2X-C dataset in Figure 5. The latency results in a delay of point clusters from road infrastructure (marked by yellow dots) compared to the point clusters from the ego car (marked by blue dots) at the current timestamp. Consequently, the ego car erroneously detects

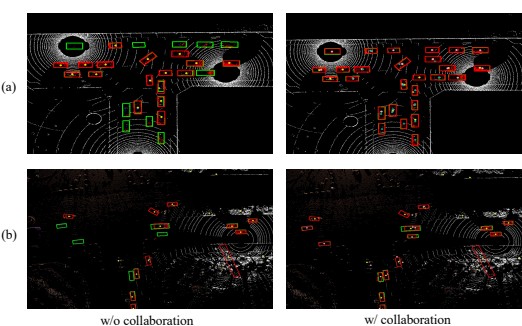

Figure 4: Qualitative comparison results of our **CPPC** with and without collaboration on the test sets of (a) V2XSet and (b) DAIR-V2X-C, respectively. The green bounding boxes represent the ground-truth, and the red ones depict our predictions.

them as distinct objects, resulting in an escalation of false-positive predictions. After refinement by our latency compensation module, the delayed point clusters can be adjusted to the correct positions, enhancing the detection results. Further, we illustrate the detection results before and after pose correction on the test set of the V2XSet dataset in Figure 6. By aligning clusters among all agents, we can correct the noisy poses and obtain bounding boxes with high precision.

## 5 RELATED WORK

### 5.1 COLLABORATIVE PERCEPTION

Collaborative perception can be systematically classified into three primary types based on distinct fusion stages: early Chen et al. (2019b); Arnold et al. (2020), intermediate, and late Shi et al. (2022);

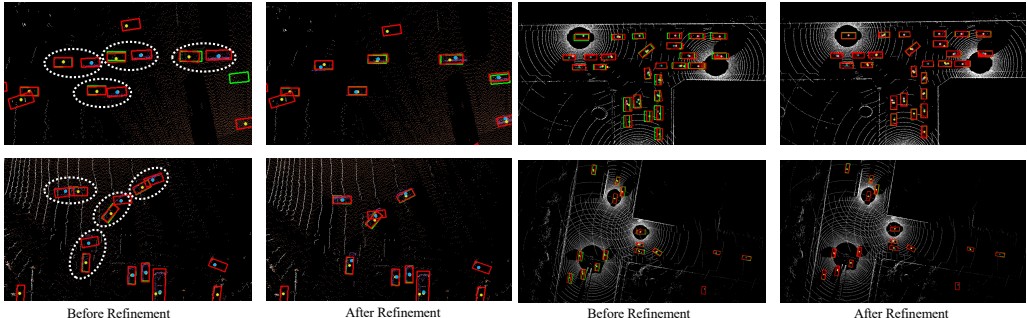

Figure 5: Qualitative comparison results of our **CPPC** before and after latency compensation.

Figure 6: Qualitative comparison results before and after pose correction.

Zeng et al. (2020); Rauch et al. (2012); Glaser & Kira (2023) collaboration. We focus on intermediate collaboration in this paper, which enables the exchange of intermediate features created by the involved agents and has demonstrated significant potential in recent years. Considering the bandwidth constraints in practical application scenarios, various cooperation strategies are proposed to decide who Liu et al. (2020b), when Liu et al. (2020a), and where Hu et al. (2022); Wang et al. (2023b) to communicate. After receiving features from other agents, existing methods adopt attention mechanism Xu et al. (2022b;a; 2023a); Zhang et al. (2022a); Wang et al. (2023a); Yang et al. (2023), graph neural network Wang et al. (2020); Li et al. (2021a), maxout Bai et al. (2022); Guo et al. (2021); Qiao & Zulkernine (2023), and addition Marvasti et al. (2020) to aggregate complementary scene information. This paper proposes a novel collaborative message unit named point cluster and demonstrates its superiority over the BEV map for collaborative perception.

### 5.2 SPARSE DETECTORS

To address quantization errors caused by voxelization, point-based detectors Shi et al. (2019); Yang et al. (2020); Qi et al. (2019); Zhang et al. (2022b); Qi et al. (2017b); Fan et al. (2022); Chen et al. (2023); Huang et al. (2023); Fu et al. (2024) have emerged as a popular research topic. PointR-CNN Shi et al. (2019) is recognized as groundbreaking research in the advancement of this line of work. Taking inspiration from Hough voting, VoteNet Qi et al. (2019) initially casts votes for object centroids and subsequently generates high-quality proposals based on the voted center. FSD Fan et al. (2022) is the pioneering fully sparse 3D object detector, which treats instances as groups and gets rid of the dependence on the neighborhood query. In this paper, we study the shortcomings of BEV-based collaboration methods and propose the point cluster as the collaborative message unit, which keeps the low-level structure information to support effective and efficient collaboration.

## 6 CONCLUSION AND DISCUSSION

In this paper, we concentrate on the multi-agent collaborative perception task, illustrating that current approaches are hampered by the inherent limitations of existing message units leading to sub-optimal collaboration. To handle this issue, we create a brand new message unit for collaborative perception, namely point cluster, and based on this we further present a novel collaborative framework **CPPC**. The core idea of **CPPC** involves representing scenes through object-level point clusters, which are sparse and encompass comprehensive information about objects. These clusters can be compressed efficiently without losing geometric or high-level object information and can be integrated through set matching and merging. To deal with time latency and pose errors encountered in real environments, we align point clusters from spatial and temporal dimensions and propose parameter-free solutions for them. Extensive experiments on two collaborative perception benchmarks show our method outperforms previous state-of-the-art methods.

**Limitation and future work.** By leveraging the low-level information within point clusters, we introduce parameter-free solutions to enhance the robustness of our approach. Despite good results, we may need to manually adjust hyperparameters when facing a new environment. In the future, we plan to address this issue using novel techniques specifically designed for adapting to environmental changes. In addition, we also plan to extend our **CPPC** to handle tasks considering temporal modeling, like tracking and forecasting Yu et al. (2023).

ACKNOWLEDGMENTS

This research is supported in part by National Key R&D Program of China (2022ZD0115502), National Natural Science Foundation of China (NO. 62461160308, U23B2010), "Pionee" and "Leading Goose" R&D Program of Zhejiang (No. 2024C01161).

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

## A METHOD DETAILS

### A.1 ALGORITHM: SEMTANIC AND DISTRIBUTION GUIDED FARTHEST POINT SAMPLING

### A.2 POINT CLUSTER DECODING

We denote the intermediate representation of point clusters in the scene after aggregation as $\ddot{M} = \{\ddot{m}^s\}_{s=1}^{N_{object}}$, where $N_{object} = \|M_{unique}\| + \|M_{share}\|$ is the potential object number in the scene observed by all involved agents. Different from BEV-based message aggregation methods, $\ddot{P}^s$ contains complete low-level structure information, which can be utilized to enhance the precision of the proposal bounding box $\ddot{B}$. In detail, we apply an additional SIR module that includes $L_2$ layers on $\ddot{M}$, which predicts the box residual $\Delta_{res}$ to its corresponding ground truth box. For each point cluster $\ddot{m}^s$, we generate its point feature $\ddot{F}_{point}^s$ by concatenating its offsets from the cluster proposal $\ddot{B}^s$ and the cluster feature $\ddot{F}^s$. This introduces the proposal boundary information to the SIR module, which can handle the size ambiguity problem to a certain extent Li et al. (2021b). We define the residual loss $\mathcal{L}_{res}$ as the L1 distance between $\Delta_{res}$ and the ground-truth residual $\hat{\Delta}_{res}$. In addition, we define the soft classification label as $\min(1, \max(0, 2u - 0.5))$ following previous works Shi et al. (2020a;b), where $u$ is the 3D Intersection-of-Union (IoU) between the predicted proposal and the ground truth. We adopt cross-entropy loss as the IoU loss $\mathcal{L}_{iou}$. Considering all losses in our framework, the total training loss $\mathcal{L}$ can be formulated as:

$$\mathcal{L} = \mathcal{L}_{seg} + \mathcal{L}_{center} + \mathcal{L}_{reg} + \mathcal{L}_{cls} + \mathcal{L}_{res} + \mathcal{L}_{iou}. \tag{9}$$

---

**Algorithm 1** Semantic and Distribution guided Farthest Point Sampling Algorithm. $N_{\text{point}}$ is the number of input points and $N_{\text{sample}} = N_{\text{point}} \times \zeta$ is the number of sampled points controlled by a predefined sampling rate $\zeta$.

---

**Input:** coordinates $\boldsymbol{P} = \{\boldsymbol{p}^1, \ldots, \boldsymbol{p}^{N_{\text{fg}}}\} \in \mathbb{R}^{N_{\text{point}} \times 3}$;
       semantic scores $\boldsymbol{S}_{\text{f}} = \{\boldsymbol{s}_{\text{f}}^1, \ldots, \boldsymbol{s}_{\text{f}}^{N_{\text{point}}}\} \in \mathbb{R}^{N_{\text{point}}}$;
       distribution scores $\boldsymbol{S}_{\text{d}} = \{\boldsymbol{s}_{\text{d}}^1, \ldots, \boldsymbol{s}_{\text{d}}^{N_{\text{point}}}\} \in \mathbb{R}^{N_{\text{point}}}$.
**Output:** sampled key point set $\widetilde{\boldsymbol{P}} = \{\widetilde{\boldsymbol{p}}^1, \ldots, \widetilde{\boldsymbol{p}}^{N_{\text{sample}}}\}$

1: initialize an empty sampling point set $\widetilde{\boldsymbol{P}}$
2: initialize a distance array $\boldsymbol{D}_{\text{point}}$ of length $N_{\text{point}}$ with all $+\infty$
3: initialize a visit array $\boldsymbol{V}$ of length $N_{\text{point}}$ with all zeros
4: **for** $n = 1$ **to** $N_{\text{sample}}$ **do**
5:      **if** $n = 1$ **then**
6:          $o = \arg\max(\{(\boldsymbol{s}_{\text{f}}^k)^{\lambda_{\text{s}}} \cdot (\boldsymbol{s}_{\text{d}}^k)^{\lambda_{\text{d}}} | \boldsymbol{V}^k = 0\}_{k=1}^{N_{\text{point}}})$
7:      **else**
8:          $\widetilde{\boldsymbol{D}}_{\text{point}} = \{(\boldsymbol{s}_{\text{f}}^k)^{\lambda_{\text{s}}} \cdot (\boldsymbol{s}_{\text{d}}^k)^{\lambda_{\text{d}}} \cdot \boldsymbol{d}_{\text{point}}^k | \boldsymbol{V}^k = 0\}_{k=1}^{N_{\text{point}}}$
9:          $o = \arg\max(\widetilde{\boldsymbol{D}}_{\text{point}})$
10:     **end if**
11:     add $\boldsymbol{P}^o$ to $\widetilde{\boldsymbol{P}}$, $\boldsymbol{V}^o = 1$
12:     **for** $u = 1$ **to** $N$ **do**
13:        $\boldsymbol{d}_{\text{point}}^u = \min(\boldsymbol{d}_{\text{point}}^u, \|\boldsymbol{p}^u - \boldsymbol{p}^o\|)$
14:     **end for**
15: **end for**
16: **return** $\widetilde{\boldsymbol{P}}$

---

## B  DATASETS

*DAIR-V2X-C* Yu et al. (2022) is the first to provide a large-scale collection of real-world scenarios for vehicle-infrastructure collaborative autonomous driving. It contains 38,845 frames of point cloud data annotated with almost 464k 3D bounding boxes representing objects in 10 different classes. Since the original DAIR-V2X-C does not include objects beyond the camera's view, we have adopted the complemented annotations encompassing the 360-degree detection range, which are relabeled by Hu *et al.* Hu et al. (2022).

*V2XSet* Xu et al. (2022a) is a large-scale V2X perception dataset founded on CARLA Dosovitskiy et al. (2017) and OpenCDA Xu et al. (2021), which explicitly takes into account real-world noises like localization error and transmission latency. V2XSet has 11,447 frames (6,694/ 1,920/2,833 for train/validation/test respectively) captured in 55 representative simulation scenes that cover the most common driving scenarios in real life. Each scene typically involves 2-7 agents engaged in collaborative perception.

*OPV2V* Xu et al. (2022b) is a vehicle-to-vehicle collaborative perception dataset, cosimulated by OpenCDA Xu et al. (2021) and Carla Dosovitskiy et al. (2017), which includes 12K frames of 3D LiDAR point clouds and RGB images with 230K annotated 3D boxes.

## C  EXPERIMENTS

### C.1  DATASET STATISTICS.

### C.2  COMMUNICATION VOLUME.

In real-world applications, collaborative perception methods must achieve a delicate equilibrium between communication volume and precision due to the typically limited and variable communication bandwidth. The communication volume is calculated as follows:

$$\text{Comm} = \log_2(N \times C \times 16/8), \tag{10}$$

where $N$ represents the number of collaborative message units, $C$ represents the number of channels, and the data is transmitted in fp16 data type, resulting in minimal performance impact. The volume in bits is then converted to bytes using the logarithm base 2.

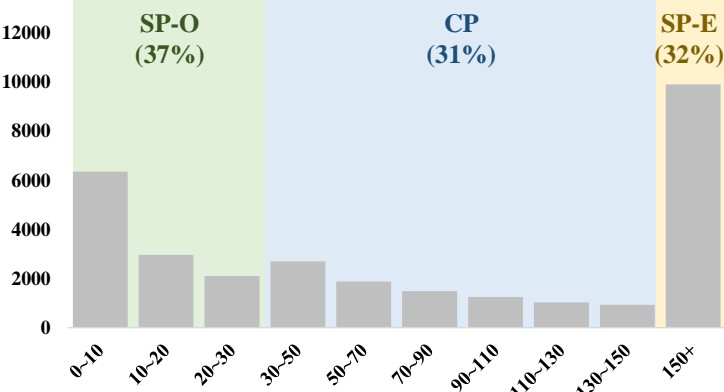

Figure 7: Illustration of histogram of all targets and proportions of those belonging to SP-O, SP-E, and CP categories in the test set of DAIR-V2X-C.

### C.3    COMPUTATIONAL COMPLEXITY OF POINT CLUSTER AGGREGATION

The computational complexity discussed primarily aims to demonstrate the efficiency of message aggregation based on sparse point clusters in comparison to dense BEV map representations. The computational complexity of message aggregation based on point clusters is solely related to the number of potential objects, which is significantly smaller than the scale of the quadratic relationship with the collaboration scope.

Assume there are $N_{\text{agent}}$ agents in the scene, so each tuple in $\boldsymbol{M}_{\text{share}}$ contains at most $N_{\text{agent}}$ matched point clusters. According to eq (5) and eq (6), the computational complexity of aggregating single tuple in $\boldsymbol{M}_{\text{share}}$ is $\mathcal{O}(\max_{1 \leq i \leq N_{\text{agent}}} |\boldsymbol{P}^i| + N_{\text{agent}} + C N_{\text{agent}} + N_{\text{agent}})$, where $\boldsymbol{P}^i$ is the set of points in point cluster and and $C$ is the channel number of cluster features. After point sampling during point cluster packing, $|\boldsymbol{P}^i|$ can be reduced significantly as shown in Table 4. And Table 3 shows that $C$ can also be reduced to a small constant. Overall, the computational complexity of aggregating single tuple in $\boldsymbol{M}_{\text{share}}$ can be reduced to $\mathcal{O}(D)$, where $D$ is a constant. Assume there are $N_{\text{object}}$ possible objects in the scene, there are at most $N_{\text{object}}$ tuples in $\boldsymbol{M}_{\text{share}}$, while $\boldsymbol{M}_{\text{unique}} = \emptyset$. The computational complexity of aggregating all tuples is $\mathcal{O}(D N_{\text{object}})$.

### C.4    ADDITIONAL ABLATION STUDIES

Table 7: Features for object pose calculation during pose correction. "0.1" label of each column means experiments with standard deviation 0.1 for both heading error (°) and positional error ($m$).

| Feature | Pose Error | | | | |
|---|---|---|---|---|---|
| | 0.1 | 0.2 | 0.3 | 0.4 | 0.5 |
| Point Center | 75.13/45.44 | 76.89/46.59 | 78.50/48.20 | 79.60/51.03 | 78.90/52.07 |
| Cluster Center | 90.22/86.95 | 90.34/87.01 | 90.04/87.06 | 90.10/87.06 | 89.97/86.94 |

**Features for Object Pose Calculation during Pose Correction.**   We evaluate AP@0.5/AP@0.7 of different features for object pose calculation during pose correction in Table 7. "Point Center" denotes representing the object pose with the mean coordinates of cluster points. "Cluster Center" denotes representing the object pose with the estimated cluster center. The experiments demonstrate that utilizing the "Point Center" method for determining the object's pose correction results in notable performance deterioration. Due to Lidar typically scanning objects partially, directly representing a point cluster by averaging the coordinates of all cluster points can lead to a significant offset from the true object center.

**Filtering background information.**   We evaluate the recall of points belonging to target objects after segmentation. The results demonstrate that segmentation can recall more than 90% of target object points. This finding confirms that point clusters generated from segmentation results can effectively represent potential object parts, enabling the fusion module to complete object information utilizing information from all agents and achieve better performance.

Table 8: Time cost ($ms$) analysis.

| V2X-ViT | Total | CPPC | | | Decoder | Others |
|---|---|---|---|---|---|---|
| | | PCE | PCP | PCA | | |
| 153.31 | 79.52 | 53.71 | 3.21 | 9.69 | 12.70 | 0.21 |

Table 9: Different point-based operators.

| | AP@0.5 | AP@0.7 |
|---|---|---|
| PointNet | 89.67 | 85.82 |
| DGCNN | 91.39 | 87.68 |
| Meta-Kernel | 91.52 | 88.48 |
| SIR | **92.01** | **89.99** |

**Inference time cost analysis.** We evaluate the time cost of our method's entire pipeline and compare it with one of the state-of-the-art methods in Table 8, i.e., V2X-ViT. Comparing the first and second columns, the results show that our method significantly reduces the time cost compared with previous collaborative sensing methods, indicating the efficiency of communication with our new message unit point cluster. Moreover, the collaborative modules (**PCP** and **PCA**) occupy a relatively small portion of our entire pipeline, which verifies collaboration based on point cluster can impose little additional burden on existing single-vehicle detectors.

**Different point-based operators.** We evaluate AP@70 of different existing point-based operators on V2XSet in Table 9, where SIR achieves the best performance.

## C.5 QUALITATIVE COMPARISON

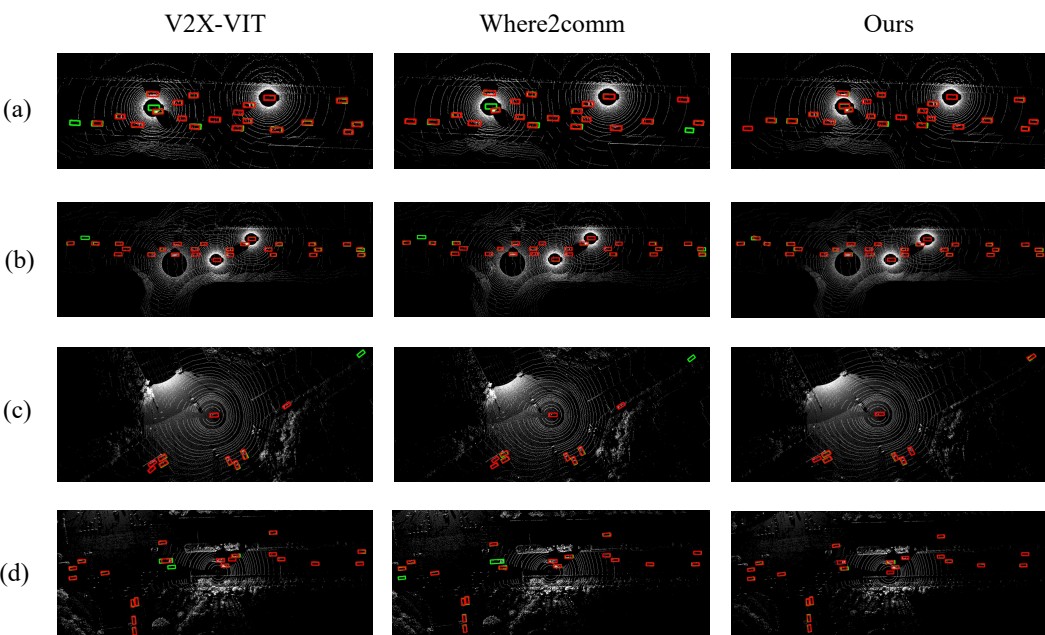

Figure 8: Qualitative comparison results of our CPPC with state-of-the-art methods on V2XSet (a, b) and DAIR-V2X-C (c, d). The green bounding boxes represent the ground-truth, and the red ones depict predictions.

