# OpenReview forum: "Point Cluster: A Compact Message Unit for Communication-Efficient Collaborative Perception"
_ICLR.cc/2025/Conference — ICLR 2025 Poster_

### Official Review · Reviewer_4p6F · 2024-10-21

**Soundness:** 3
**Presentation:** 3
**Contribution:** 3
**Rating:** 6
**Confidence:** 2

**Summary:**

This paper proposes to use Point Cluster as the message unit for collaborative perception. Previous intermediate methods use BEV map as the message unit, which suffer from weak object features, inefficient message aggregation and vague boundary. The proposed point cluster-based framework can solve these problems well. Extensive experiments on V2XSet, OPV2V, and DAIR-V2X-C demonstrate the effectiveness of the proposed method.

**Strengths:**

1. Using point cluster as an intermediate message unit is novel and well-motivated.
2. The proposed CPPC framework, as well as the PCE, PCP and PCA modules are solid.
3. State-of-the-art performance on mainstream benchmarks.

**Weaknesses:**

My main concerns are about the effectiveness and efficiency of the proposed method, for which I think more ablation studies are required.
1. Ablation studies on PCP, PCA, pose correction and latency compensation are required.
2. The authors claim the BEV representation is inefficient during aggregation. Is there any comparison between BEV and Point Cluster-based methods?

**Questions:**

In section 3.5, can the authors detail "spatial cluster matching"? If the pose between two agents is not accurate, how to match point clusters from different agents into a single object?

---

> ### Author Response · Authors · 2024-11-20
>
> Thank you for your valuable comments and kind words to our work. Below we address specific questions.
>
> **Q1: Ablation studies on PCP, PCA, pose correction and latency compensation.**
> - **About PCP.** Without the PCP module, which achieves message packaging for other agents, this method degrades to single-agent perception for the ego agent without messages from other agents. We evaluate this baseline on the validation set of V2XSet, achieving AP\@0.7 of 67.00, which is far behind our full CPPC that achieves AP\@0.7 of 89.99. These results demonstrate the limitations of single-agent perception and validate the necessity of collaboration for achieving more complete scene perception.
> - **About PCA.** Without the PCA module, which is designed to aggregate point cluster features from multiple agents, we directly merge boxes in messages from multiple agents like late collaboration. This setting is evaluated on the validation set of V2XSet, achieving AP\@0.7 of 77.88, which is far behind our full CPPC that achieves AP\@0.7 of 89.99. This validates that our PCA module can effectively utilize rich object information.
> - **About pose correction.** We evaluate our CPPC under a positional error of 0.6m and a heading error of 0.6$^\circ$ without the pose correction module, achieving an AP\@0.7 of 34.22 on the V2XSet validation set. This result is significantly lower than the AP\@0.7 of 87.37 achieved with the pose correction module, highlighting its critical role in enhancing performance.
> - **About latency compensation.** We evaluate our CPPC under time latency 500ms without the latency compensation module, achieving an AP\@0.7 of 62.08 on the test set of DAIR-V2X-C. This result is below than the AP\@0.7 of 64.61 achieved with the latency compensation module.
>
> **Q2: Comparison with BEV-based methods during aggregation.**
> - In Appendix C.2, we theoretically analyze that the computational complexity of point cluster aggregation is approximately linearly related to the number of potential targets in the scene, i.e., $\mathcal{O}(DN_\text{object})$, where $D$ is a constant. However, the computational complexity for BEV-based message aggregation is $\Omega(HW)$, which is related to the square of the perception range. Since $N_\text{object} \ll HW$ in most real scenes, our point cluster-based method is more efficient for long-range aggregation. We evaluate the inference time cost of our PCA module, i.e., 9.69 ms, and aggregation module in V2X-ViT, i.e., 41.42 ms.
>
> **Q3: Spatial cluster matching when pose errors exist.**
> - **Details of spatial cluster matching.** We calculate the Euclidean distance between the centers of point clusters detected by different agents. If the distance between two point cluster centers is less than a specified threshold $\epsilon_\text{pose}$, they are considered to belong to the same potential object.
> - **Cluster matching under pose errors.** When pose errors are present, aligning the coordinate space of other agents with the ego agent may result in misaligned point clusters. To address this issue, we first obtain an initial matching result through spatial cluster matching. Subsequently, we optimize Eq. (7) to estimate the accurate poses of the surrounding agents, thereby correcting pose errors to achieve more reasonable matching based on the refined poses. The experiments depicted in Fig. 4 (c) and (d) demonstrate that the proposed strategy performs effectively under various pose error scenarios.
> - **Extension work.** We appreciate your thoughtful suggestion and have carefully considered this aspect of the method. Indeed, in scenarios with significant pose errors—though rare—it becomes challenging for a manually set threshold to address all cases. To enhance the robustness of spatial cluster matching, we extend our current approach to a maximum common subgraph (MCS)-based spatial matching [1]. In this method, each cluster mapping within the MCS represents point clusters corresponding to the same potential object across the cluster graphs of different agents.
>
> [1] A Partitioning Algorithm for Maximum Common Subgraph Problems, IJCAI, 2017.

---

> > ### Comment · Reviewer_4p6F · 2024-11-24
> >
> > Thanks for the authors' reply. It solves my problems and I do not have any other questions.

---

### Official Review · Reviewer_xrYN · 2024-11-04

**Soundness:** 4
**Presentation:** 4
**Contribution:** 3
**Rating:** 6
**Confidence:** 3

**Summary:**

This paper introduces a communication-efficient collaborative perception (CPPC) framework for vehicle-to-everything autonomous driving. Unlike previous methods that mainly rely on BEV features, the CPPC framework leverages point clusters to control its computational complexity and alleviate issues like high-level information loss. Specifically, the CPPC framework consists of a point cluster picking module, a pose alignment and latency compensation module, and a point cluster aggregation module to generate cluster features for subsequent predictions. The CPPC framework outperforms existing BEV-based methods on three public datasets, including V2XSet, OPV2V, and DAIR-V2X-C.

**Strengths:**

\+ This paper is written well and the motivation behind the proposed method is easy to follow.

\+ The proposed method is a systematic solution for multi-agent perception, which exploits point clusters as the intermediate representation with controllable communication costs, to balance efficiency and effectiveness.

\+ The proposed method outperforms several previous methods on three public datasets consistently. The extensive experimental results validate the effectiveness of the proposed modules.

**Weaknesses:**

\- There are four thresholds used in the proposed method, which may be hard to choose in complex or unseen scenarios.

\- The proposed method mainly considers BEV-based methods for comparisons, however, it is unclear how significant its technical contributions are compared with previous point-cloud-based methods (e.g., Cooper, F-cooper). I am concerned about this because the techniques used in the main components of the proposed method, including point cluster picking, pose correction, and the SD-FPS, are quite common in the research communities of point cloud processing.

**Questions:**

My current concerns are mainly about the technical contributions of the proposed method. I may change my rating after reading other reviewers' comments and the authors' rebuttal. Here are some questions I wish could be addressed:

Q1: In the proposed SD-FPS method, how to balance the weights of semantic and spatial distances ($\lambda_s$ and $\lambda_d$)? How will they affect the performance of the proposed method?

Q2: Does the proposed method perform well in a crowded environment where many objects are exhibited?

---

> ### Author Response · Authors · 2024-11-20
>
> Thank you for your valuable comments and kind words to our work. Below we address specific questions.
>
> **Q1: Thresholds may be hard to choose in complex or unseen scenarios.**
> - This is a great question. We ablate the thresholds involved in our CPPC in Tables 5 and 6. The results show minimal performance fluctuations over a wide range, demonstrating that our method is robust to threshold variations. Furthermore, after determining these thresholds through ablation experiments on the validation set of the V2XSet dataset, we directly trained and tested on the OPV2V dataset under the same threshold settings, achieving state-of-the-art performances. This validates the generalization ability of our method.
>
> **Q2: Comparision with previous point-cloud-based methods.**
> - **CPPC vs. Cooper.** Cooper is a foundational early collaboration method. Through cooperative sensing, Cooper can merge and align the shared data that is collected from nearby vehicles, which may provide data scopes coming from different positions and angles. Cooper relies on three types of human-designed ROI-based rules to select point clouds to meet actual bandwidth constraints, which may not generalize to complex real scenerios like DAIR-V2X-C. Differently, our CPPC directly transmits sparse point clusters to the ego agent, which achieves the state-of-the-art performance-bandwidth balance.
> - **CPPC vs. F-Cooper.** Both CPPC and F-Cooper are intermediate collaboration methods. Constrained by its voxel-based representation and RPN network, F-Cooper must expand the feature map size during the information aggregation stage to cover the perception range of different agents. This leads to unnecessary computations when collaborative agents are far apart. In contrast, the computational complexity of point cluster aggregation in CPPC depends solely on the number of potential objects, effectively avoiding this issue. We evaluate F-Cooper on the V2XSet and DAIR-V2X-C datasets, achieving an AP\@0.7 of 87.06 on the V2XSet test set and 60.31 on the DAIR-V2X-C test set. In comparison, CPPC outperforms F-Cooper with absolute improvements of 2.49 and 9.08 in AP\@0.7, respectively.
> - **Novelty of implementation techniques.** The core contribution of our paper is the introduction of a novel point cluster-based communication paradigm for collaborative perception. Building on the concept of point clusters, we develop a comprehensive system, CPPC, which achieves a state-of-the-art trade-off between performance and bandwidth, alongside robustness to various noisies. To implement our approach, we enhance existing technologies (e.g., FPS) due to their wide applicability within the point cloud processing community. Crucially, it is the design of the point cluster that enables the integration of these technologies into the domain of collaborative perception—an advancement that was not feasible with previous methods. We will explore novel implementation techniques which can further improve our CPPC in the future.
>
> **Q3: Ablation of the weights of semantic and spatial distances.**
> - We evaluate different weights, $\lambda_s$ and $\lambda_d$, for semantic and density scores, respectively, with a sample ratio of 1/16 on the test set of DAIR-V2X-C. For simplicity, we set $\lambda_d=1-\lambda_s$. Best performances are achieved when $\lambda_d=0.4, \lambda_s=0.6 $.
> | **Method** | 0.0   | 0.2   | 0.4   | 0.6   | 0.8   | 1.0   |
> |------------|-------|-------|-------|-------|-------|-------|
> | **AP\@0.5** | 76.14 | 76.28 | 76.41 | 76.47 | 76.32 | 76.16 |
> | **AP\@0.7** | 69.01 | 69.07 | 69.22 | 69.03 | 68.84 | 68.62 |
>
> **Q4: Crowded environment performances.**
> - This is a great question. We analyze the target count in DAIR-V2X-C, as crowded environments typically involve a large number of agents. The dataset includes 42 scenarios with 30 or more targets. Our CPPC achieves an AP\@0.5 of 70.83 and an AP\@0.7 of 63.45, both surpassing the overall performance of existing methods, demonstrating its robustness in crowded environments.

---

> > ### Comment · Reviewer_xrYN · 2024-11-26
> >
> > I appreciate the reply from the authors and my concerns have been addressed. Hence, I have decided to maintain my rating.

---

### Official Review · Reviewer_1x7n · 2024-11-04

**Soundness:** 4
**Presentation:** 4
**Contribution:** 4
**Rating:** 8
**Confidence:** 4

**Summary:**

This paper proposes a novel message exchange unit called point cluster for collaborative perception. Point clusters can efficiently and compactly represent an object's location, structure, and semantic information. The proposed CPPC framework includes point cluster-based encoding, packing, exchange, and integration methods. CPPC improves both communication efficiency and perception accuracy while being robust to various real-world noises. Experimental results on various benchmarks demonstrate that CPPC significantly outperforms existing methods.

**Strengths:**

This paper systematically analyzes the advantages and disadvantages of existing collaborative perception methods (early, intermediate, late collaboration) and proposes a rational solution considering the inherent trade-off between communication efficiency and perception performance.
The proposed point cluster is a compact message unit that selectively combines the strengths of existing methods and efficiently represents both structural and semantic information of objects.
CPPC introduces parameter-free approaches to handle pose error and time latency issues encountered in real-world scenarios, ensuring robustness to various noise levels.
This paper clearly identifies the current challenges and limitations in the field of collaborative perception and demonstrates how the proposed method can rationally solve them.
Through clear problem formulation, the collaborative perception problem is mathematically defined, and based on this, the motivation and principles of the proposed method are lucidly explained.

**Weaknesses:**

Based on the proposed technique, the amount of data that can be transmitted is limited. Therefore, it is crucial to prioritize foreground objects and bring them in a sparse manner to distinguish the overall shape, rather than bringing only a few parts. Regarding this aspect, I wonder if there are any criteria or tendencies for determining which points should be selected and what rules should be followed. Additionally, I am curious about the extent of performance degradation when background information is included instead of solely focusing on the foreground.

If we follow the logic mentioned above, I wonder if we can expect sufficient performance improvement in early or late collaboration by only bringing foreground information and adjusting it according to the available bandwidth.

- In Figure 4(a), there is a section where the performance of the proposed method rises sharply. I am curious about the reason behind this phenomenon and how the performance of the proposed method would differ from existing technologies if the communication volume is lower than this point.
- On Page 7, Line 373, it is mentioned that the proposed method shows improvements of 5.7%, 7.3%, and 12.8%, but it is unclear which methods are being compared.
- On Page 7, Line 377, the term "late collaboration" is used. I am curious about which specific technique this refers to.

**Questions:**

If we assume some objects are moving at high speeds, I think the errors caused by delay would vary for each object. It seems this aspect might not be covered, does the proposed method handle this problem?

---

> ### Author Response · Authors · 2024-11-20
>
> Thank you for your valuable comments and kind words to our work. Below we address specific questions.
>
> **Q1: Criteria for determining which points should be selected.**
> - As shown in Algorithm 1 (Appendix A.1), our proposed SD-FPS strategy prioritizes points in point clusters with clear semantic features or key structural information. Concretely, we measure the ambiguous of each point by its confidence score $s_f$ in segmentation head of PCE, where a higher $s_f$ indicates richer semantic information for distinguishing objects. As for structural measures, the distribution density score $s_d$ is inversely proportional to its density estimation: $\frac{1}{\mathcal{N}(p)}\sum_{q\in\mathcal{N}(p)}K(p,q)$, where $\mathcal{N}(p)$ is a point set in the nearby area of $p$ , and $K(\cdot,\cdot)$ is a gaussian kernel function to measure the similarity between position of two points. Thus, a lower $s_d$ indicates removing it will not significantly affect the object shape.
>
> **Q2: Performance degradation when background information is included.**
> - We include all foreground and background points by setting the foreground threshold in PCE to zero. Experimental results on the V2XSet validation set indicate an AP\@0.5 of 91.37 and an AP\@0.7 of 89.23, reflecting absolute decreases of 0.64 and 0.76, respectively. The primary cause of this performance degradation is the introduction of background points, which can distort the shape description of potential objects during clustering. However, the negative impact is relatively minor, as the PCP phase retains only point clusters with positive proposals (line 229 in paper). Notably, while the inclusion of background points has limited impact on performance, it substantially increases communication bandwidth requirements.
>
> **Q3: Performance analysis in early or late collaboration by only bringing foreground information.**
> - Early collaboration is expected to achieve the highest performance when bandwidth constraints are not a factor, as it avoids information loss during communication. We agree that transmitting only foreground point clouds is an effective strategy for early collaboration methods to maintain accuracy while reducing bandwidth usage. To validate this, based on the early collaboration method that directly aggregates all raw point clouds, we introduce our trained foreground point segmentation network into the fusion pipeline. By applying an appropriate threshold to filter foreground points, we reduce the communication volume of the early collaboration method to a similar magnitude as other methods, enabling a fair comparison.
>
> | **Method**                                  | **AP\@0.5** | **AP\@0.7** | **Communication Volume** |
> |:---------------------------------------------|:------------:|:------------:|:--------------------------:|
> | V2X-ViT                                     | 67.44      | 51.87      | 15.62                    |
> | Where2comm                                  | 71.22      | 60.22      | 15.66                    |
> | Early collaboration w/ foreground filtering | 75.01      | 65.71      | 15.32                    |
> | CPPC                                        | 76.89      | 69.39      | 15.46                    |
>
> - On the DAIR-V2X-C test set, the early collaboration method with foreground filtering achieves 75.01 AP\@0.5 and 65.71 AP\@0.7, significantly outperforming Where2comm and V2XViT under the similar communication volume. Although the early collaboration with foreground filtering demonstrates a certain level of competitiveness, our approach still exhibited significantly superior accuracy, achieving performance improvements of 1.88 in AP\@0.5 and 3.68 in AP\@0.7. This improvement is attributed to the introduction of the point cluster feature and the more reasonable keypoint selection strategy employed in SD-FPS. In contrast, later collaboration methods use bounding boxes as message units, which primarily contain only foreground information.

---

> ### Author Response · Authors · 2024-11-20
>
> **Q4: Sharply rises in Figure 4 (a).**
> - This is a good question. When the communication volume is limited to approximately 14, the performance of our CPPC drops sharply. We believe that under such a stringent communication bandwidth constraint, the sampled points in the information packaging stage become overly sparse. Consequently, the remaining key points fail to adequately represent the shape characteristics of potential objects, making it challenging to generate high-quality detection boxes.
> - When the communication volume is 14, we can calculate the spatial confidence mask rate in the state-of-the-art method, i.e., Where2comm, according to eq (10) in appendix C.1. The number of allowed collaborative message units is calculated: $N=\frac{2^{Comm}\times8}{16\times C}=\frac{2^{14}\times8}{16\times256}=32$, indicating mask rate is 0.9948 for BEV feature maps with shape $48\times 128$. This means that only around 0.5\% BEV features are allowed to be transmitted, which are almost equivalent to no collaboration, with 55.54 AP\@0.7 on the test set of DAIR-V2X-C.
> - Overall, even under such strict communication volume constraints, our CPPC achieves 64.32 AP\@0.7, outperforming Where2comm by an absolute margin of 8.78 in AP\@0.7. This result validates the superiority of our point cluster-based communication mechanism.
>
> **Q5: It is unclear which method the accuracy improvement results are compared with.**
> - On each dataset, we compare our approach with the current best-performing methods: OPV2V on V2XSet, CoAlign on OPV2V, and V2X-ViT on DAIR-V2X-C.
>
> **Q6: Technique of late collaboration method.**
> - Late collaboration approaches adopt bounding boxes as message units. We implement this late collaboration baseline by directly using $\ddot{B}^s$ for point clusters matching the same potential objects, i.e., in $M_\text{share}$, and $B$ for objects exclusively observed by a single agent, i.e., in $M_\text{unique}$, as the final outputs, respectively. Thus, the late collaboration can not benefit from completing object semantic and structure information from mutli resources in PCA and use PCD (Appendix A.2) to refine them.
>
> **Q7: Solution for delay-induced errors that vary by object.**
> - This is a great question. In this paper, we evaluate the robustness to time latency on the test set of DAIR-V2X-C, which includes 10 km of city roads, 10 km of highway, 28 intersections, and 38 km² of driving regions, encompassing diverse weather and lighting conditions from real-world scenarios. Therefore, we believe this evaluation is sufficiently comprehensive to validate the effectiveness of our latency compensation module.
> - However, we acknowledge that our existing solution has limitations. For instance, when objects move at high speeds, the artificially defined upper and lower bounds for temporal cluster matching may struggle to handle such significant displacements. We appreciate you highlighting this issue and propose addressing it by introducing point cluster flow prediction to mining temporal associations in cluster feature space in future works.

---

> > ### Comment · Reviewer_1x7n · 2024-11-24
> >
> > Thank you for the authors' response. All concerns about the methodology and weaknesses I raised have been addressed. However, the experiment section is still difficult to read. When discussing performance improvements, the baseline methods for comparison is not directly specified. For its clarity and quick comprehension, I encourage the authors to revise this section.

---

> > > ### Author Response · Authors · 2024-11-24
> > >
> > > Thank you for your suggestion. We have updated the draft and uploaded it, with the changes highlighted in blue. The updates are as follows:
> > > 1. The second-highest performance accuracy is now highlighted in blue in Table 1. Additionally, we specify the comparison method in the text description (lines 372–376).
> > > 2. A detailed description of the late collaboration baseline has been added (lines 377–378).

---

### Official Review · Reviewer_tujs · 2024-11-12

**Soundness:** 2
**Presentation:** 3
**Contribution:** 2
**Rating:** 6
**Confidence:** 1

**Summary:**

This paper presents a new message unit, the "point cluster," to improve collaborative perception efficiency in multi-agent systems. Unlike existing message units such as raw point clouds, bounding boxes, or BEV maps, the point cluster format minimizes bandwidth usage while retaining essential structural and semantic information. Representing objects with point coordinates, a cluster center, and semantic features, this approach allows efficient inter-agent information exchange, enhances object alignment, and preserves object structure for more accurate detection. A new framework, CPPC, combines point packing and aggregation modules, addressing issues like bandwidth constraints, time delay, and pose errors, and achieves state-of-the-art performance on several benchmarks.

**Strengths:**

1. **Bandwidth Efficiency**: The proposed "point cluster" message unit significantly reduces communication bandwidth by capturing only essential foreground object information in a sparse format, making it highly efficient compared to dense representations like raw point clouds and BEV maps.

2. **Enhanced Object Detection Accuracy**: By preserving detailed structural and semantic information, the point cluster improves object detection accuracy, especially in complex multi-agent scenarios, demonstrating superior performance on established collaborative perception benchmarks.

3. **Robustness to Real-world Challenges**: The CPPC framework includes robust mechanisms for handling pose errors and time delays, crucial for real-world applications. Parameter-free solutions for pose and latency issues make it adaptable to various levels of noise without additional tuning.

4. **Clear and Effective Writing**: The paper is well-written, with a clear explanation of the proposed methods and thorough descriptions of experiments, which makes the complex technical content accessible and supports the credibility and reproducibility of the research findings.

**Weaknesses:**

In terms of methodology, there is a noticeable reliance on FSD, which slightly reduces the originality. However, overall, the approach is still reasonable.

**Questions:**

No questions.

---

> ### Author Response · Authors · 2024-11-20
>
> Thank you for your valuable comments and kind words to our work. Below we address specific questions.
>
> **Q1: Relationship with FSD.**
> - **Single-agent perception vs. multi-agent collaborative perception.** The goal of single-agent perception is to enhance perceptual capabilities within the scope of the ego view. In contrast, multi-agent collaborative perception seeks to address the occlusion limitation inherent to single-agent systems by facilitating complementary information exchange between surrounding agents. A key challenge in multi-agent collaborative perception is achieving superior scene-level perception performance under constrained communication costs. The development of multi-agent approaches must address challenges such as packaging observation information in a bandwidth-efficient manner, effectively aggregating multi-source data, and ensuring robustness against issues like pose errors and time latency—challenges absent in single-agent approaches.
> - **FSD vs. CPPC.** We agree that both our CPPC and FSD both utilize sparse point cloud representations for 3D object detection instead of dense BEV representations. However, the contributions differ fundamentally:
>   - For single-agent perception, FSD innovatively proposes a fully sparse pipeline, addressing the issue of center feature missing and eliminating the need for time-consuming neighborhood queries in purely point-based methods. It achieves state-of-the-art performance in 3D object detection while being much faster than previous detectors, especially in long-range settings. Although FSD has greatly promoted the development of point cloud-based sparse detectors, it did not consider the communication-related issues faced by multi-agent collaborative perception.
>   - For multi-agent collaborative perception, CPPC introduces a novel communication paradigm based on our proposed message unit, i.e., point cluster, which effectively addresses key challenges in existing BEV-based intermediate collaboration approaches. These challenges include object feature degradation during message packing, inefficient message aggregation for long-range collaboration, and the communication of implicit structural representations. First, by prioritizing semantically and structurally rich points in point clusters through our proposed SD-FPS strategy, CPPC achieves the state-of-the-art performance-bandwidth balance. Second, the computational complexity of point cluster aggregation scales efficiently with the number of potential objects in the scene, rather than increasing quadratically with the perception range, making it highly suitable for long-range collaboration. Furthermore, extensive experiments demonstrate that CPPC maintains robustness to a wide range of pose errors and time latency without additional fine-tuning, thanks to the explicitly preserved coordinate information within the point clusters.

---

> > ### Comment · Reviewer_tujs · 2024-11-27
> > **Response to author rebuttal**
> >
> > The responses of authors address most of my concerns. I will maintain my rating.

---

### Meta-Review · Area_Chair_Rwvx · 2024-12-21

**Metareview:**

This paper looks at the problem of collaborative perception, taking the perspective of minimizing size (and hence bandwidth) while retaining essential structural and semantic information. To do this, the Point Cluster representation is proposed, which represents objects via point coordinates, a cluster center, and point features. This representation is integrated into a new framework for packing, aggregating, and decoding messages for transmission. Results are shown across a number of collaborative perception benchmarks, demonstrating improved results over a number of competing methods.

  Reviewers appreciated the perspective of early, intermediate, and late collaboration as well as the method's improved bandwidth efficiency, performance, and and clarity of the writing. The addition of experiments with respect to real-world robustness, including pose errors and time latency, was also mentioned. A number of weaknesses were raised by reviewers, including methodological similarity to existing methods both in this (Cooper, F-Cooper) and other fields (FSD), unexplained phenomena in some of the graphs (e.g. performance spikes) as well as overall lack of clarity in the experiment section, lack of ablations, and sensitivity to the hyper-parameters. The authors provided a comprehensive rebuttal, including additional results especially analyzing early vs. late collaboration and relationship to foreground information. Reviewers expressed that this rebuttal satisfied most of their concerns, and all recommended acceptance.

  Based on this, I recommend acceptance of this paper. Overall, the paper provides a nice perspective of levels/types of collaboration and an interesting method that balances compression and retention of relevant information, both structural and semantic. I encourage the authors to incorporate all of the new discussions and results in the main paper.

**Additional Comments On Reviewer Discussion:**

Reviewers raised a number of concerns, and authors provided a comprehensive rebuttal including additional results/analysis. All reviewers mentioned that the rebuttal addressed their concerns and all recommended acceptance.

---

### Decision · Program_Chairs · 2025-01-22

Accept (Poster)